# Urinary Mercury Levels and Predictors of Exposure among a Group of Italian Children

**DOI:** 10.3390/ijerph17249225

**Published:** 2020-12-10

**Authors:** Maria Luisa Astolfi, Matteo Vitali, Elisabetta Marconi, Stefano Martellucci, Vincenzo Mattei, Silvia Canepari, Carmela Protano

**Affiliations:** 1Department of Chemistry, “Sapienza” University of Rome, 00185 Rome, Italy; silvia.canepari@uniroma1.it; 2Department of Public Health and Infectious Diseases, “Sapienza” University of Rome, 00185 Rome, Italy; matteo.vitali@uniroma1.it (M.V.); elisabetta.marconi@uniroma1.it (E.M.); carmela.protano@uniroma1.it (C.P.); 3Biomedicine and Advanced Technologies Rieti Center, “Sabina Universitas”, 02100 Rieti, Italy; stefano.martellucci@uniroma1.it (S.M.); vincenzo.mattei@uniroma1.it (V.M.)

**Keywords:** pediatric age, non-invasive matrix, biomonitoring, toxic element, air pollution, cold vapor atomic fluorescence spectrometry

## Abstract

Urinary mercury (Hg) levels are suitable to assess long-term exposure to both elemental and inorganic Hg. In this study, the urinary Hg levels of 250 children (aged 6–11 years) from three areas with different anthropogenic impacts in the Rieti province, central Italy, were assessed. The Hg concentrations were in the range of 0.04–2.18 µg L^−1^ with a geometric mean equal to 0.18 µg L^−1^ [95% confidence interval (CI), 0.17–0.20 µg L^−1^] or 0.21 µg g^−1^ creatinine (95% CI, 0.19–0.23 µg g^−1^ creatinine), and a reference value calculated as 95th percentile of 0.53 µg L^−1^ (95% CI, 0.44–0.73 µg L^−1^) or 0.55 µg g^−1^ creatinine (95% CI, 0.50–0.83 µg g^−1^ creatinine). In all cases, urinary Hg data were below the HBM-I values (7 µg L^−1^ or 5 µg g^−1^ creatinine) established for urine, while the 95th percentile was above the German Human Biomonitoring Commission’s RV95 (0.4 µg L^−1^) set for children without amalgam fillings. A significant correlation (*p* < 0.05) was found between creatinine-corrected results and residence area, with higher urinary Hg levels in children living in the industrial area. Multiple linear regression analysis showed that creatinine was the main predictor of urinary Hg.

## 1. Introduction

Mercury (Hg) is a toxic heavy metal for both environmental and human health [1,2]. It can be released into the environment from natural and anthropogenic sources [3,4,5]. Hg can be found in the environment mainly in the following three chemical forms: organic (e.g., methyl- and ethyl-Hg), inorganic (e.g., mercuric chloride), and elemental or metallic [6,7]. Exposure to all Hg chemical forms may occur via inhalation, ingestion, or dermal contact [8,9]. Exposure to inorganic and elemental Hg occurs mainly through inhalation of ambient air [10]. However, the use of consumer products containing Hg, such as hair treatment products, skin lightening creams, and soaps, can also be an additional source of inorganic Hg intake [11,12]. Instead, dietary habits and fish consumption are the main routes of methyl-Hg intake [13].

The health impact of Hg exposure in humans varies widely and depends on several factors such as the Hg chemical form, dose, duration of exposure, and the susceptibility of the exposed subject [1,12,14,15]. Organic Hg may cause auditory, visual, and sensory problems and adverse health effects on nervous and cardiovascular systems [3,16], while inorganic Hg is mainly associated with negative outcomes on central nervous system function and kidneys [4,17]. Because of their susceptibility, children, neonates, and fetuses are subpopulations at higher risk of health effects due to Hg intake [18,19,20]. Human biomonitoring (HBM) can provide useful information on the exposure and internal dose of chemicals within the general population, various subpopulations, and individuals by analyzing biomarkers in bodily fluids and tissues, such as blood, serum, urine, breast milk, hair, and nails [21,22,23,24,25,26,27]. In particular, urinary total Hg levels are suitable to assess long-term exposure to both elemental and inorganic Hg [28,29,30,31,32,33]. Besides, urinary Hg levels require creatinine adjustment to account for renal function and hydration differences [6,34]. For this reason, only urine samples with creatinine levels in the range from 0.3 to 3.0 g L^−1^ should be considered to determine reference values for a specific population and to perform toxicological investigations [35,36]. The German Environmental Agency defined two guidance values (HBM I = 7 µg L^−1^ or 5 µg g^−1^ creatinine and HBM II = 25 µg L^−1^ or 20 µg g^−1^ creatinine) for the interpretation of Hg urinary concentrations [37,38]. Therefore, exposure profiles and specific reference values for different groups are needed to evaluate and compare biomonitoring data on Hg exposure in different populations, subgroups, or individuals [15]. Both exposure profiles and reference values should be periodically revised to consider changes in environmental contamination levels and population exposure [39].

In this work, a biomonitoring study was conducted to obtain the Hg exposure profiles among a group of Italian school-aged children by measuring total urinary Hg levels and collecting possible influencing factors via a questionnaire [gender, age, body mass index (BMI), sport activity, fish consumption, smoking habits of cohabitant smoker(s), residence area] on its variability.

To date, given the scientific evidence on Hg toxicity in children, the most susceptible group of the population, it is essential to provide Hg exposure data during pediatric age currently lacking worldwide. Besides, to our knowledge, no data are available on this issue for the Italian school-aged population.

## 2. Materials and Methods

### 2.1. Design, Study Population, and Urine Sample Collection

A cross-sectional study was conducted in June 2018 in Rieti province (Latium, central Italy) (Figure 1). A total of 369 children aged 5–11 years were recruited from primary district schools, as described previously [40]. 

One spot urine sample was collected for each participant (details are reported by Antonucci et al. (2020) [40]), subdivided into aliquots, and frozen at −20 °C until analysis. Urine samples with creatinine concentration <0.3 or >3.0 g L^−1^ or when the amount was not sufficient were excluded. Moreover, results on participants with at least one parent not Italian were excluded from data processing, as ethnicity influences body metabolism and excretion [41]. Thus, statistical elaborations were performed on 250 samples, as summarized in Table 1.

Figure 1 reports the three studied areas, located in Rieti province, central Italy, and named Poggio Moiano, Rieti, and Cittaducale. The Rieti province is characterized by a low urbanization rate, as described previously [42]. The three studied areas were selected based on some urbanization indicators; in particular, the population density [inhabitants (IN)/km^2^] and the green land consumption percentage were equal to 226.4 IN/km^2^ and 7.2% in Rieti, 104.3 IN/km^2^ and 5% in Cittaducale and 92.6 IN/km^2^ and 4.9% in Poggio Moiano [43,44]. In addition, Cittaducale presents a large industrial center for photovoltaic cell processing, currently not working. Therefore, although Poggio Moiano and Cittaducale have similar characteristics in urbanization, the first was considered a predominantly rural site, while the second an industrial area. Rieti was considered a predominantly urban area.

Both children’s parents signed informed consent to participate in the study. For each participant, detailed information on gender, age, weight and height, residence area, health status, sport activity, fish consumption, and parents’ smoking habits and educational level were recorded in a questionnaire reported in an already published paper [40]. In particular, exposure to environmental tobacco smoke was considered when at least one smoker lived with the child. Sport activity was investigated, assessing the participation in at least one organized sport out of the school hours and if this activity was outdoor or indoor. Fish consumption was considered only if fish-based food were present in the last meals before sampling. The study protocol and the questionnaire were reviewed and approved by the local Ethical Committee (Policlinico Umberto I, “Sapienza” University of Rome; protocol code 2894). All data obtained from biological samples and questionnaires were treated anonymously and used only for scientific purposes.

### 2.2. Chemical Analysis

Total Hg concentrations in urine were determined by cold vapor atomic fluorescence spectrometry (CVAFS; AFS 8220 Titan, FullTech Instruments, Rome, Italy) with 5% HCl (36%; Promochem, LGC Standards GmbH, Wesel, Germany) as a carrier, and 0.05% NaBH4 (Sigma-Aldrich, St. Louis, USA) in 0.05% NaOH (98%, anhydrous pellets, RPE for analysis, ACS–ISO; Carlo Erba Reagents, Milan, Italy) as a reducing agent. All solutions were prepared using deionized water (resistivity, ≤18.3 MΩ cm) generated by an Arioso Power I RO-UP Scholar UV deionizer (Human Corporation, Songpa-Ku, Seoul, Korea). The instrumental conditions are presented in previous studies [45,46]. Standard Hg solutions for CVAFS calibration range from 0.04 to 1.5 μg L^−1^ were prepared by diluting adequate stock standard volumes (1002 ± 7 mg L^−1^; SCP Science, Baie D’Urfé, Canada) in 3% HCl. A minimum R2 value of 0.999 was considered, and the linearity range was verified by Mandel’s fitting test [47]. Urine samples were diluted 5-fold with 3% (*v*/*v*) HCl. Urinary Hg concentrations were calculated as both μg L^−1^ of urine and μg g^−1^ of creatinine to normalize individual variation due to each participating child’s differing hydration states. Urine samples that presented critical creatinine values lower than 0.3 g L^−1^ or higher than 3 g L^−1^ were excluded from the Hg analysis [35,36,48]. The method detection limit (MDL) of Hg in this study was 0.03 μg L^−1^. The analytical procedure was tested using certified reference material for urine (Seronorm™ Trace Elements Urine L-2 LOT 1403081) supplied by Sero AS (Billingstad, Norway). Intra- and inter-day precision calculated as the coefficient of variation percentage was 1.8 and 5.5%, respectively, and the trueness bias percentage was 6%. A standard solution (0.4 μg L^−1^) and blanks (3% HCl) were checked daily to control instrumental drift, cross-contamination, and memory effects.

### 2.3. Statistical Elaboration

Statistical analysis was performed using MedCalc version 19.5.3 (MedCalc Software Ltd., Ostend, Belgium) and SPSS version 25 for Windows (IBM Corp., Armonk, NY, USA). To describe urinary Hg concentration in samples, descriptive statistics (arithmetic and geometric means, standard deviation, minimum, maximum, and percentiles) were calculated together with their corresponding 95% confidence intervals (95% CI), as recommended by the International Federation of Clinical Chemists [49]. All data were above the MDL and not normally distributed [D’Agostino-Pearson test for normal distribution; coefficient Skewness = 4.520, *p* < 0.0001 and coefficient of Kurtosis = 30.27, *p* < 0.0001 for unadjusted (μg L^−1^) values; and coefficient Skewness = 2.751, *p* < 0.0001, and coefficient of Kurtosis = 11.76, *p* < 0.0001 for normalized to creatinine (μg g^−1^ creatinine) values]. Hence non-parametric tests, such as the Mann–Whitney, Kruskal–Wallis and Spearman correlation tests, were used to study the association between urinary Hg level and sociodemographic/fish consumption characteristics of children. A *p*-value less than 0.05 (two tailed) was regarded as significant. Some studied variables were coded before statistical processing in accordance with the following information obtained via questionnaire: gender (0 = male; 1 = female); age (0 = 6–8 years old; 1 = 9–11 years old); geographical location (0 = rural area; 1 = urban area; 2 = industrial area); ponderal status based on sex-specific body mass index-for-age growth charts elaborated by the Centers for Disease Control and Prevention [50] (underweight and healthy weight = 0 or overweight and obese = 1); sport activity (0 = no; 1 = indoor; 2 = outdoor); exposure to environmental tobacco smoke (0 = no; 1 = yes); university degree study of at least one parent (0 = no; 1 = yes); fish consumption before sampling (0 = no; 1 = yes).

A backward multiple linear regression was used to assess the influence of the predictor variables, including creatinine levels, as recommended by Barr et al. (2005) [51], on urinary Hg levels. The *p*-values used as the stay and entry criterion were 0.10 and 0.05, respectively. Linear regression coefficients were calculated with their 95% confidence intervals to indicate the uncertainty and precision of the sample statistical estimates [52,53]. Residual plots checked the influence of possible outliers and regression assumptions [54].

## 3. Results

Table 1 shows the characteristics of 250 children (136 boys and 114 girls) participating in the study. 45.6% of children aged 6–8 years and 54.4% were 9–11 year-old. Most children (50%) lived in the industrial area, practiced indoor sports (53.2%), and did not consume fish (82.4%). Almost a third of the children were overweight or obese. A similar proportion concerning exposure to environmental tobacco smoke (ETS) emerged in the monitoring campaign (41.6% of exposed vs. 57.2 of not exposed to ETS). A small group of children (14.8%) had at least one graduate parent.

Table 2 presents the distribution of urinary Hg levels in the study population. The Hg concentrations were in the range 0.04–2.18 µg L^−1^ with a geometric mean of 0.18 µg L^−1^ [95% confidence interval (CI), 0.17–0.20 µg L^−1^], which is similar to the creatinine-corrected mean of 0.21 µg g^−1^ creatinine (95% CI, 0.19–0.23 µg g^−1^ creatinine), and a value calculated as 95th percentile of 0.53 µg L^−1^ (95% CI, 0.44–0.73 µg L^−1^) or 0.55 µg g^−1^ creatinine (0.50–0.83 µg g^−1^ creatinine). 

Figure 2 compares the geometric mean concentrations in this study with those from other children’s relevant biomonitoring researches. The average concentration obtained in our research was similar to that obtained in children populations without dental amalgam of Germany (0.19 μg L^−1^) [55], and lower than those reported by other authors [54,56,57,58,59]. However, the urinary Hg levels were about twice those found in children from Germany in 2007 [37].

Table 3 shows the influence of the studied variables on the urinary Hg levels using the non-parametric statistical tests. The Mann–Whitney or Kruskal–Wallis test was used to compare the urinary Hg level according to the qualitative variables analyzed. The Spearman correlation test was performed to study the relationship between the urinary Hg levels and the quantitative variables selected. A significant correlation (*p* < 0.05) between Hg level and geographical residence area of children was found only in creatinine-corrected results, with higher levels in children living in the industrial area. Creatinine concentration mean was 0.97 ± 0.42 g L^−1^ and showed a significantly (*p* = 0.0001) positive correlation with urinary Hg levels. No other selected variables influenced Hg urinary excretion.

Table 4 presents the multiple linear regression analysis results indicating the possible association between urinary Hg concentration and independent variables. Creatinine concentration was the most important predictor of exposure to the urinary Hg level and geographical location (grouped as rural and urban as control) entered in the model with a *p*-value = 0.089.

## 4. Discussion

### 4.1. Urinary Hg Concentration

Metal concentrations are used in health-risk assessments and to compare populations with different environmental exposures. Age is often not considered by some clinical laboratories when interpreting urinary analysis, and the concentration intervals calculated on the adult population were used even in the case of the exposure evaluation of children [60]. However, adult reference values may not be protective enough for children because children cannot be considered small adults [26,59,61]. Children are considered the most vulnerable subgroups of the population to environmental contaminants and exhibit different behavior than adults regarding excretion and accumulation of metals due to the difference in their immature detoxification mechanisms, physiology, body growth, lifestyle, and physical activity [9,15,62,63]. In addition, children exposed to low levels of trace metals, even below concentrations considered safe for the general population, showed neurobehavioral and cognitive changes [64].

We calculated urinary Hg concentration also as the 95th percentile with the 95% CI (Table 2). Although these values cannot indicate Hg exposure’s hazardous levels, they can be statistically considered to describe the Hg body burden [37]. 

The urinary Hg was mainly related to elemental and inorganic Hg exposure and most clearly associated with dental fillings [55,65,66], but unfortunately, we did not address this information in the present study. However, the Italian Ministry of Health in 2014 recommended avoiding the use of dental amalgam in children under six years old, pregnant or breastfeeding women, and in patients with severe kidney disease or with an allergy to the material itself [67]. The European Union (EU) with the Regulation (EU) 2017/852 has adopted a ban, starting from 1 July 2018, on the use of amalgam fillings in children under fifteen, in pregnant or breastfeeding women [68]. The action that the European Parliament and the Council approved is part of a broader implementation of the Minamata Convention’s objectives [69], aimed at limiting the use and release of Hg into the environment. The geometric mean found in our study (0.18 µg L^−1^) was similar to the value of 0.19 µg L^−1^ reported by Pesch et al. (2002) [55] in children without amalgam fillings aged 8–10 years but lower than the geometric means reported in other biomonitoring studies in children [54,56,57,58,59], as shown in Figure 1. In all cases, our urinary Hg data were below the HBM-I values (7 µg L^−1^ or 5 µg g^−1^ creatinine) established for urine by Schulz et al., (2011) [38], while the 95th percentile [0.53 µg L^−1^ (95% CI, 0.44–0.73 µg L^−1^)] (Table 2) was above the German Human Biomonitoring Commission’s RV95 (0.4 µg L^−1^) set for children without amalgam fillings [38,70].

### 4.2. Determinants of Hg Exposure 

A potential source of inorganic Hg may be diet, vaccines, and the use of cosmetics (such as soaps and skin-lightening creams) that can contain Hg salts [11,71]. Some studies showed that fish consumption might also influence the urinary Hg levels in children without amalgam fillings [14,54,72,73]. A study by Sherman et al., (2013) [30] shows using naturally occurring Hg stable isotopes that urinary Hg can also derive from organic Hg in fish due to internal de-methylation. In contrast, there are no significant differences between those who consume fish or not in our study. This may be because we only considered the meals before the urine sampling and not the children’s eating habits. The urinary Hg levels are often linked to gender-related differences [56]. In general, females appear to have a greater ability to excrete Hg in the urine than males [74]. The higher concentration of Hg in females’ urine than males is also shown in our study, but the difference between urinary Hg concentrations in males (geometric mean, 0.19 µg g^−1^ creatinine) and females (geometric mean, 0.23 µg g^−1^ creatinine) is not significant. As Woods et al. (2007) [74], the urinary Hg concentration does not necessarily represent an increased risk of Hg toxicity than in males. The use of metabolic biomarkers and toxicokinetic studies may help correctly interpret urinary Hg data to assess safe Hg exposure levels [74]. Significant differences (*p* = 0.010) were obtained among the three geographical areas studied with higher urinary Hg levels in children living in the industrial area [0.24 µg g^−1^ creatinine (95% CI, 0.21–0.27 µg g^−1^ creatinine)]. This result was also confirmed by regression analysis: geographical location entered in the model with a *p*-value at the limit of significance). Unfortunately, environmental data on air Hg concentrations are not available, thus it is not possible to give conclusive explanations, but it is plausible that the industrial area presents greater airborne levels of Hg. The use of heavy metals, including Hg, is widespread in the industry, including that for the manufacture of solar photovoltaic cells [75,76,77]. Although lower or comparable to the urinary Hg concentrations of children living in other countries of the world, our results highlight the need for further environmental monitoring studies.

The most important predictor of urinary Hg exposure among the monitored children was the urinary level creatinine (Table 4). Other authors showed a positive correlation between the urinary concentrations of some substances such as other elements [54,64], Li [26], cotinine [78], trans, trans-muconic acid, and S-phenylmercapturic acid [79] and creatinine in children. To our knowledge, this is the first time that this correlation was also evidenced for urinary Hg levels, adding new evidence in the field of biomonitoring studies.

### 4.3. Study Limitation 

The present study has some limitations. First of all, the recruitment of participants is limited to a specific region of Italy, and the sample size, although large, does not define general reference values for urinary Hg. However, information on current Hg exposure among children in the Lazio region is valuable because, to our knowledge, this is the first study on Italian children and one of the few in the world concerning the need for scientific evidence on this issue. Finally, the present study is a cross-sectional study; therefore, it does not allow for an evaluation over time.

## 5. Conclusions

Urinary Hg data were generally lower in Italian children than Hg levels in other countries, probably due to different eating habits and environmental factors. We would like to highlight the importance of establishing updated reference values for urinary Hg concentrations in children to detect possible toxic effects due to this metal. Although our results provide important concentration data, further studies should be conducted in other Italian geographical areas and evaluate temporal trends to better interpret biomonitoring data for risk management and assessment.

## Figures and Tables

**Figure 1 ijerph-17-09225-f001:**
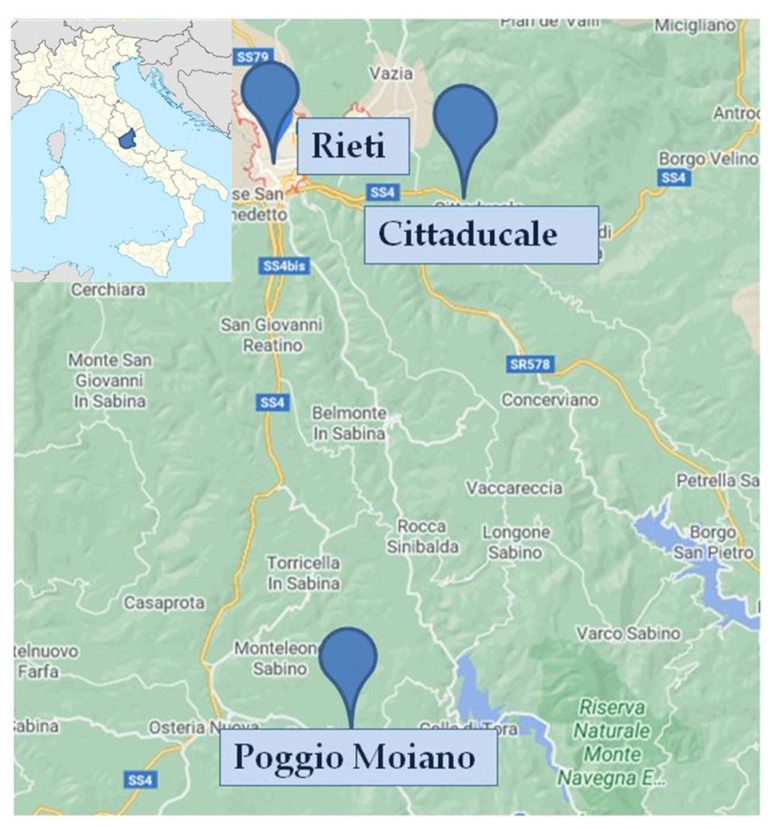
Map of the three sampling sites in the study area (province of Rieti, central Italy).

**Figure 2 ijerph-17-09225-f002:**
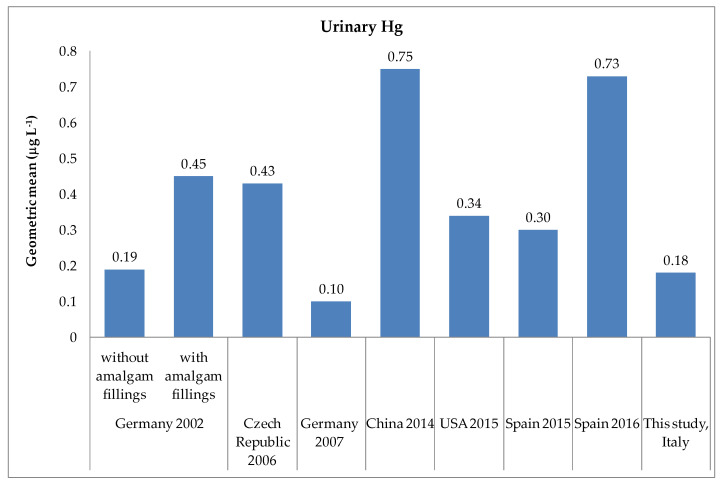
Comparison of urinary mercury concentrations (as geometric mean, µg L^−1^) obtained in our study and other biomonitoring studies in children: Germany 2002 [55], Czech Republic 2006 [56], Germany 2007 [37], China 2014 [57], USA 2015 [58], Spain 2015 [59], Spain 2016 [54], and this study, Italy.

**Table 1 ijerph-17-09225-t001:** Studied population characteristics.

Variable	*N* = 250
**Gender**	*n* (%)
Male	136 (54.4)
Female	114 (45.6)
**Age (years)**	9 (6–11) ^a^
6–8 years	114 (45.6)
9–11 years	136 (54.4)
**Height (cm)**	133 (100–165) ^a^
*Missing data*	36
**Weight (kg)**	30 (18–90) ^a^
*Missing data*	21
**BMI (kg m^−2^)**	17.5 (10.7–34.3) ^a^
*Missing data*	40
**Ponderal status**	
Underweight and healthy weight	123 (49.2)
Overweight and obese	76 (30.4)
*Missing data*	*51*
**Geographical location**	
Rural (Poggio Moiano)	55 (22.0)
Urban (Rieti)	65 (26.0)
Industrial (Cittaducale)	125 (50.0)
*Missing data*	5
**Sport activity**	
Outdoor	68 (27.2)
Indoor	133 (53.2)
No sport activity	40 (16.4)
*Missing data*	9
**Exposure to environmental tobacco smoke**	
Yes	104 (41.6)
No	143 (57.2)
*Missing data*	3
**University degree of one of the parents**	
Yes	37 (14.8)
No	210 (84.0)
*Missing data*	3
**Fish consumption**	
Yes	44 (17.6)
No	206 (82.4)
*Missing data*	0

^a^ Values are expressed as median (minimum and maximum).

**Table 2 ijerph-17-09225-t002:** Urinary mercury (Hg) levels of children in Italy (*n* = 250).

Variable	Hg (µg L^−1^)	Hg (µg g^−1^ Creatinine)
Geometric mean (95% CI)	0.18 (0.17–0.20)	0.21 (0.19–0.23)
Arithmetic mean (95% CI)	0.24 (0.21–0.27)	0.27 (0.24–0.29)
Standard deviation	0.22	0.21
Median (95% CI)	0.18 (0.17–0.20)	0.22 (0.20–0.24)
Minimum	0.04	0.04
2.5th (95% CI)	0.04 (0.03–0.054)	0.04 (<MDL–0.054)
5th (95% CI)	0.06 (0.04–0.07)	0.06 (0.04–0.07)
25th (95% CI)	0.12 (0.11–0.14)	0.13 (0.12–0.16)
75th (95% CI)	0.28 (0.26–0.33)	0.34 (0.31–0.38)
95th (95% CI)	0.53 (0.44–0.73)	0.55 (0.50–0.83)
97.5th (95% CI)	0.73 (0.54–1.47)	0.83 (0.56–1.35)
Maximum	2.18	1.59

**Table 3 ijerph-17-09225-t003:** Association between urinary mercury (Hg) level and sociodemographic/food consumption characteristics of children.

	Hg (µg L^−1^)		Hg (µg g^−1^ Creatinine)	
Variable	GM (95% CI)	*p*-Value	GM (95% CI)	*p*-Value
**Gender**		0.474 ^a^		0.082 ^a^
Male	0.19 (0.16–0.20)		0.19 (0.17–0.22)	
Female	0.19 (0.16–0.22)		0.23 (0.20–0.26)	
**Age (years)**	−0.052 (−0.179–0.076) ^b^	0.426 ^c^	−0.0842 (−0.210–0.044) ^b^	0.198 ^c^
6–8 years	0.19 (0.17–0.22)	0.545 ^a^	0.22 (0.19–0.26)	0.317 ^a^
9–11 years	0.18 (0.16–0.20)		0.20 (0.17–0.22)	
**Ponderal Status**		0.656 ^a^		0.450 ^a^
Underweight and healthy weight	0.19 (0.17–0.24)		0.21 (0.16–0.24)	
Overweight and obese	0.16 (0.15–0.22)		0.19 (0.15–0.25)	
**Geographical location**		0.192 ^d^		*0.010 ^d^*
Rural (Poggio Moiano)	0.18 (0.14–0.24)		0.18 (0.14–0.23)	
Urban (Rieti)	0.18 (0.14–0.24)		0.18 (0.14–0.23)	
Industrial (Cittaducale)	0.20 (0.18–0.22)		0.24 (0.21–0.27)	
**Sport activity**		0.978 ^d^		0.930 ^d^
Outdoor	0.18 (0.15–0.21)		0.20 (0.17–0.25)	
Indoor	0.18 (0.16–0.21)		0.21 (0.19–0.24)	
No sport activity	0.19 (0.16–0.24)		0.23 (0.19–0.28)	
**Exposure to environmental tobacco smoke**		0.135 ^a^		0.095 ^a^
Yes	0.19 (0.17–0.21)		0.22 (0.19–0.25)	
No	0.17 (0.15–0.19)		0.19 (0.17–0.22)	
**University degree of one of the parents**		0.469 ^a^		0.717 ^a^
Yes	0.17 (0.13–0.21)		0.20 (0.15–0.25)	
No	0.18 (0.16–0.20)		0.20 (0.18–0.22)	
**Fish consumption**		0.696 ^a^		0.615 ^a^
Yes	0.18 (0.14–0.22)		0.22 (0.18–0.28)	
No	0.18 (0.17–0.20)		0.21 (0.19–0.23)	
**Creatinine**	0.276 (0.156–0.389) ^b^	*0.0001 ^c^*	-	-

GM: geometric mean. The number in italic indicates a significant association among the variables considered (*p*-value < 0.05). ^a^ Mann–Whitney test. ^b^ Spearman correlation (95% confidence interval). ^c^ Spearman correlation test. ^d^ Kruskal–Wallis test.

**Table 4 ijerph-17-09225-t004:** Results of the multiple linear regression analysis.

Variables	B	Standard Error	β	*p*-Value	R^2^	Adj R^2^
Constant	−2.199	0.129		<0.001	0.070	0.062
Creatinine	0.434	0.109	0.255	<0.001		
Geographical location	0.156	0.092	0.109	0.089		

B = standardized regression coefficient; β = no standardized regression coefficient; R^2^ = squared multiple correlation of predictor variable; Adj R^2^ = adjusted squared multiple correlation of predictor variable. Variables included in the model (backward method): gender (male = 0, female = 1), age (as continuous variable), creatinine (as a continuous variable), geographical location (as rural/urban = 0, industrial = 1).

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
