# Peer review of "Urinary Mercury Levels and Predictors of Exposure among a Group of Italian Children"

_ijerph, 2020, doi:10.3390/ijerph17249225_

Round 1

Reviewer 1 Report

The authors substantially improved the manuscript and addressed sufficiently all main concerns.

It might be advisable to include the association of creatinine-corrected Hg mentioned in Line 171-173: “…geographical residence area of children was found only in creatinine-corrected results, with higher levels in children living in the industrial area…” in the abstract.

Author Response

Reviewer 1 (R1)

Comments and Suggestions for Authors

The authors substantially improved the manuscript and addressed sufficiently all main concerns.

Authors (A): Thank you very much.

R1: It might be advisable to include the association of creatinine-corrected Hg mentioned in Line 171-173: “…geographical residence area of children was found only in creatinine-corrected results, with higher levels in children living in the industrial area…” in the abstract.

A: Done, thank you for your suggestion.

Reviewer 2 Report

The manuscript has been significantly improved after authours’ revision. So, I recommend that this manuscript can be accepted after minor revision. There is only one revision suggestion:

(1)、In the line 71, the Figure1(b) can be illustrated in Figure1(a) as a small one.

Author Response

Reviewer 2 (R2)

Comments and Suggestions for Authors

The manuscript has been significantly improved after authours’ revision. So, I recommend that this manuscript can be accepted after minor revision.

Authors (A): Thank you very much.

R2: There is only one revision suggestion:

(1)、In the line 71, the Figure1(b) can be illustrated in Figure1(a) as a small one.

A: Done. Thank you for your suggestion.

This manuscript is a resubmission of an earlier submission. The following is a list of the peer review reports and author responses from that submission.

Round 1

Reviewer 1 Report

In this cross-sectional study the urinary Hg concentration of 250 children aged between 6-11 were determined with the aim to provide first reference values of children living in Italy. The children were recruited from three different areas of Italy (rural, urban, industrial) and information on relevant confounder such as gender, fish consumption, and parental smoking were provided by questionnaires.

The authors present an interesting study on reference values for urinary mercury in school-aged children. Conventional studies for reference values should be sufficiently large to represent the background exposure of the general population. Although, the information on the current mercury exposure among children of the Lazio region is valuable, higher number of participants are necessary for valid reference values. Since the recruitment is restricted to a specific region of Italy and a single metal, the data set might be of limited relevance for the general readership or responsible authorities.

Minor:

More detailed information on the question defining sport activity, environmental tobacco smoke exposure or fish consumption in the method section is desirable.

A questionnaire in Italian language might not be useful for the general readership.

Reviewer 2 Report

In this paper, the authors investigated the urinary Hg levels of 250 children from three areas in the Italy and discussed various influencing factors using multivariate statistical methods. 

Experimental scheme and statistical analysis of this article are feasible and expression logic is clear. However, the discussion and analysis in this paper are not thorough enough. The main flaws are listed as following:
1、Scientific problems and research objectives are unclear. If the authours want to determine the “Reference values” and “the main predictor of urinary Hg”, I think the authors should discuss the problem using more words. If the authours want to determine the main influencing factors of urinary Hg, I think the authors should introduce the difference of three areas in the Lazio region in detail and combining with the local actual situation to discuss the influencing factors, and put forward prevention actions or improvement measures.
2、 The article lacks innovation. The authours only tested and statistical analyzed the urinary Hg levels of 250 children. As the authours said “there is no data available on the occurrence of urinary Hg concentrations in a children's population from Italy”, perhaps this is the innovation of this article.
  In short, this paper has a certain amount of workload, but the research objectives need to be further condensed, and the analysis and discussion sections need to be further improved.